# FAN1 interaction with ubiquitylated PCNA alleviates replication stress and preserves genomic integrity independently of BRCA2

Antonio Porro [1], Matteo Berti[1], Julia Pizzolato[1], Serena Bologna[1,3], Svenja Kaden[1], Anja Saxer[1], Yue Ma[2], Kazuo Nagasawa[2], Alessandro A. Sartori [1] & Josef Jiricny[1,4]

Interstrand cross-link (ICL) hypersensitivity is a characteristic trait of *Fanconi anemia* (FA). Although FANCD2-associated nuclease 1 (FAN1) contributes to ICL repair, *FAN1* mutations predispose to karyomegalic interstitial nephritis (KIN) and cancer rather than to FA. Thus, the biological role of FAN1 remains unclear. Because fork stalling in FAN1-deficient cells causes chromosomal instability, we reasoned that the key function of FAN1 might lie in the processing of halted replication forks. Here, we show that FAN1 contains a previously-uncharacterized PCNA interacting peptide (PIP) motif that, together with its ubiquitin-binding zinc finger (UBZ) domain, helps recruit FAN1 to ubiquitylated PCNA accumulated at stalled forks. This prevents replication fork collapse and controls their progression. Furthermore, we show that FAN1 preserves replication fork integrity by a mechanism that is distinct from BRCA2-dependent homologous recombination. Thus, targeting FAN1 activities and its interaction with ubiquitylated PCNA may offer therapeutic opportunities for treatment of *BRCA*-deficient tumors.

[1] Institute of Molecular Cancer Research of the University of Zurich and ETH Zurich, Winterthurerstrasse 190, CH-8057 Zurich, Switzerland. [2] Department of Biotechnology and Life Science, Tokyo University of Agriculture and Technology, 2-24-16 Naka-cho, Koganei-shi, Tokyo 184-8588, Japan. [3] Present address: Wellcome Trust/Cancer Research UK Gurdon Institute, Tennis Court Road, Cambridge CB2 1QN, UK. [4] Present address: Institute of Molecular Life Sciences of the University of Zurich and Institute of Biochemistry of the ETH Zurich, Otto-Stern-Weg 3, Zurich 8093, Switzerland. Antonio Porro, Alessandro A. Sartori and Josef Jiricny contributed equally to this work. Correspondence and requests for materials should be addressed to A.P. (email: porro@imcr.uzh.ch) or to A.A.S. (email: sartori@imcr.uzh.ch) or to J.J. (email: jjiricny@ethz.ch)

FAN1 is an evolutionarily-conserved protein identified in a proteomic screen for MLH1-interacting factors[1]. Near its N-terminus it contains a RAD18-like CCHC zinc finger, the so-called ubiquitin-binding zinc-finger (UBZ) domain, while a PD-(D/E)XK motif found in a superfamily of repair nucleases is situated close to its C-terminus[2–5]. FAN1 also contains a SAP domain required for DNA binding and a TPR domain that mediates inter-domain interactions[6]. Disruption of the *FAN1* gene in chicken DT40 cells[7], or knock-down of its mRNA by interfering RNA[2–5], resulted in a hypersensitivity to ICL-inducing agents[2–5, 7]. Biochemical studies revealed FAN1 to be a 5′ flap endonuclease and a 5′ to 3′ exonuclease[2–5], and structural studies helped explain its ability to unhook ICLs through catalysing a series of incisions separated by three nucleotides[6, 8, 9].

Despite this wealth of knowledge, the biological role of FAN1 remains enigmatic. Its efficient recruitment to mitomycin C (MMC)-induced DNA damage foci by ub-FANCD2 and the requirement of its UBZ domain in this process strongly implied a link between FAN1 and the FA pathway. However, this prediction was not substantiated by genetic data showing that *FAN1* mutations segregate with KIN rather than with FA[10–12]. Moreover, FAN1-deficient cells are generally less sensitive to ICL-inducing agents than FA cells and, unlike FA cells, display no growth defects under normoxic conditions[7]. Cells lacking FA proteins and FAN1 are also more sensitive to treatment with ICL-inducing agents than the single mutants[7, 11]. Thus, while the contribution of FAN1 to detoxification of exogenously-introduced ICLs is beyond doubt, this polypeptide might address either a subset of ICLs that are not processed by the FA pathway, or act in concert with it to ensure rapid and efficient removal of these highly-deleterious lesions from DNA. This latter notion is supported by recent findings demonstrating that the sensitivity of FAN1-deficient cells to MMC could be fully-rescued by stable expression of a FAN1 variant mutated in or lacking the RAD18-like UBZ domain. This demonstrated that, although FAN1 can be recruited to ICLs through interaction with ub-FANCD2, it can process them also independently of it[10, 13]. Interestingly, MMC treatment of cells stably-expressing the UBZ FAN1 mutant caused extensive chromosomal instability. This was also observed upon treatment with hydroxyurea (HU)[13] that arrests replication forks through depletion of nucleotide pools. The link of FAN1 to stalled replication fork processing was independently confirmed in another study, in which the nuclease was demonstrated to cause extensive degradation of aphidicolin-blocked replication forks in FANCD2-depleted cells[14]. This evidence suggests that FAN1 may address replication forks blocked not only by ICLs, but also in other ways.

One process that hinders the progression of replication forks is the spontaneous folding of G-rich single-stranded DNA arising during transcription or replication into G-quadruplexes (G4s), consisting of two or more stacks of guanine quartets stabilised by Hoogsteen hydrogen-bonds. It has been predicted that more than 700′000 sequences in the human genome have the potential to form G4s[15].

We therefore set out to test whether FAN1 might be involved in the processing of these structures. Here, we show that FAN1 prevents fork collapse at G4s through an interaction with ubiquitylated PCNA (ub-PCNA) mediated by a previously-uncharacterised PCNA interacting peptide (PIP) motif and the UBZ domain. These results suggest that FAN1 is a novel reader of ub-PCNA. Moreover, FAN1 enhances PCNA ubiquitylation *via* a feed-forward loop, as recently described for SPRTN/DVC1[16]. In addition, the FAN1 PIP motif and its association with ub-PCNA is indispensable for FAN1 localization upon exposure to UV radiation or hydroxyurea HU. Thus, the FAN1 interaction with ub-PCNA rather than with ub-FANCD2 ensures FAN1 recruitment to stalled replication forks. In contrast, mutation of the FAN1 PIP motif does not affect FAN1 localization to ICLs, thus unveiling a separation of function of the FAN1 protein domains in different contexts. Finally, we provide evidence that FAN1 and ub-PCNA co-operate to protect genome integrity independently of BRCA2. Taken together, our data uncover a novel function of FAN1 in the metabolism of replication stress.

## Results

### FAN1 prevents fork collapse and promotes PCNA ubiquitylation.

G-quadruplexes exist in DNA only transiently; we therefore took advantage of a compound, S2T1-6OTD, an analog of telomestatin referred to henceforth as aTMS, which has been recently shown to bind G4s selectively and with high affinity[17]. We first set out to learn whether aTMS caused genomic instability in our model system, the human osteosarcoma U2OS cells. Pulse field gel electrophoretic (PFGE) analysis of genomic DNA isolated from wild type (WT) cells treated with control siRNA against luciferase (CNTL) showed that aTMS induced readily-detectable double-strand breaks (DSBs), the number of which increased 2- to 3-fold in cells treated with FAN1 siRNA (Fig. 1a, b). Interestingly, siRNA-mediated knock-down of MUS81 (Fig. 1a), a nuclease of opposite polarity to FAN1, resulted in a significant reduction in DSB occurrence, both in the singly- and doubly-depleted cells (Fig. 1b). The latter results suggest that FAN1 alleviates the genotoxicity of aTMS by directly processing a subset of G4s, but that it also addresses intermediates generated by a MUS81-dependent pathway. In the absence of MUS81, these intermediates do not arise, which results in a substantial decrease in DSB number. This is reminiscent of the situation in MMC-treated mouse embryonic fibroblasts lacking either one or both proteins[11]. The fact that genomic DNA of aTMS-treated control cells contained readily-detectable DSBs may indicate that the pathway(s) responsible for repairing the FAN1- and MUS81-dependent incisions became saturated due to the large number of G4s induced by aTMS.

In order to learn whether FAN1 was recruited to aTMS-stabilised G4s and whether its recruitment required the functional UBZ- and/or nuclease domains, we used the "protein replacement" method[18] to generate stable U2OS cell lines, in which FAN1 was substituted by its eGFP-tagged variants. This method makes use of a stably-integrated vector containing two doxycycline (Dox)-inducible cassettes: one expressing shRNA targeting mRNA that encodes an endogenous protein of choice, and the second expressing cDNA encoding the desired protein variant, in this case FAN1 carrying the WT form of the protein, the C44A/C47A mutations that abrogate the RAD18-like zinc finger of the UBZ domain (UBZ*), or the nuclease-dead (ND) variant D960A/K977A (Supplementary Fig. 1a, b). Dox treatment of these stable cell lines leads to degradation of the mRNA encoding the endogenous FAN1 protein and to expression of the desired variant (Supplementary Fig. 1b).

As shown in Supplementary Fig. 1c, WT eGFP-FAN1 formed subnuclear foci in untreated U2OS cells, predominantly those in the S and G2 phases of the cell cycle (Supplementary Fig. 1d), as revealed by Quantitative Image-Based Cytometry (QIBC). Upon aTMS treatment, the number of these foci increased significantly (Fig. 1c–e). The eGFP-FAN1 ND variant generated substantially fewer foci, both in untreated (Supplementary Fig. 1c, d) and aTMS-treated cells (Fig. 1c–e), whereas the eGFP-FAN1 UBZ* variant failed to form distinct foci and appeared to localise to the nucleoli (Fig. 1c–e and Supplementary Fig. 1c, d). FAN1 was previously shown to be required for the successful rescue of halted replication forks[13, 14], possibly by restraining their progression and allowing for their repair[13]. In the latter study,

the forks were arrested by HU-induced dNTP depletion and we wanted to learn whether FAN1 played a similar role in the metabolism of stabilized G4s. We therefore monitored DNA synthesis and cell cycle distribution in exponentially-growing U2OS cell lines expressing eGFP-FAN1 WT, eGFP-FAN1UBZ* and eGFP-FAN1 ND by FACS analysis through incorporation of 5-ethynyl-2′-deoxyuridine (EdU). As seen in Fig. 1f (upper panel), the percentage of S phase cells in the three untreated populations was not substantially different (29–37%) and EdU incorporation was also similar. aTMS treatment caused an almost complete inhibition of EdU incorporation in the U2OS eGFP-FAN1 WT line, similarly to what was reported for HU[13], whereas the lines expressing the UBZ* and ND FAN1 variants were also affected, albeit to a substantially lesser extent (Fig. 1f, lower panel).

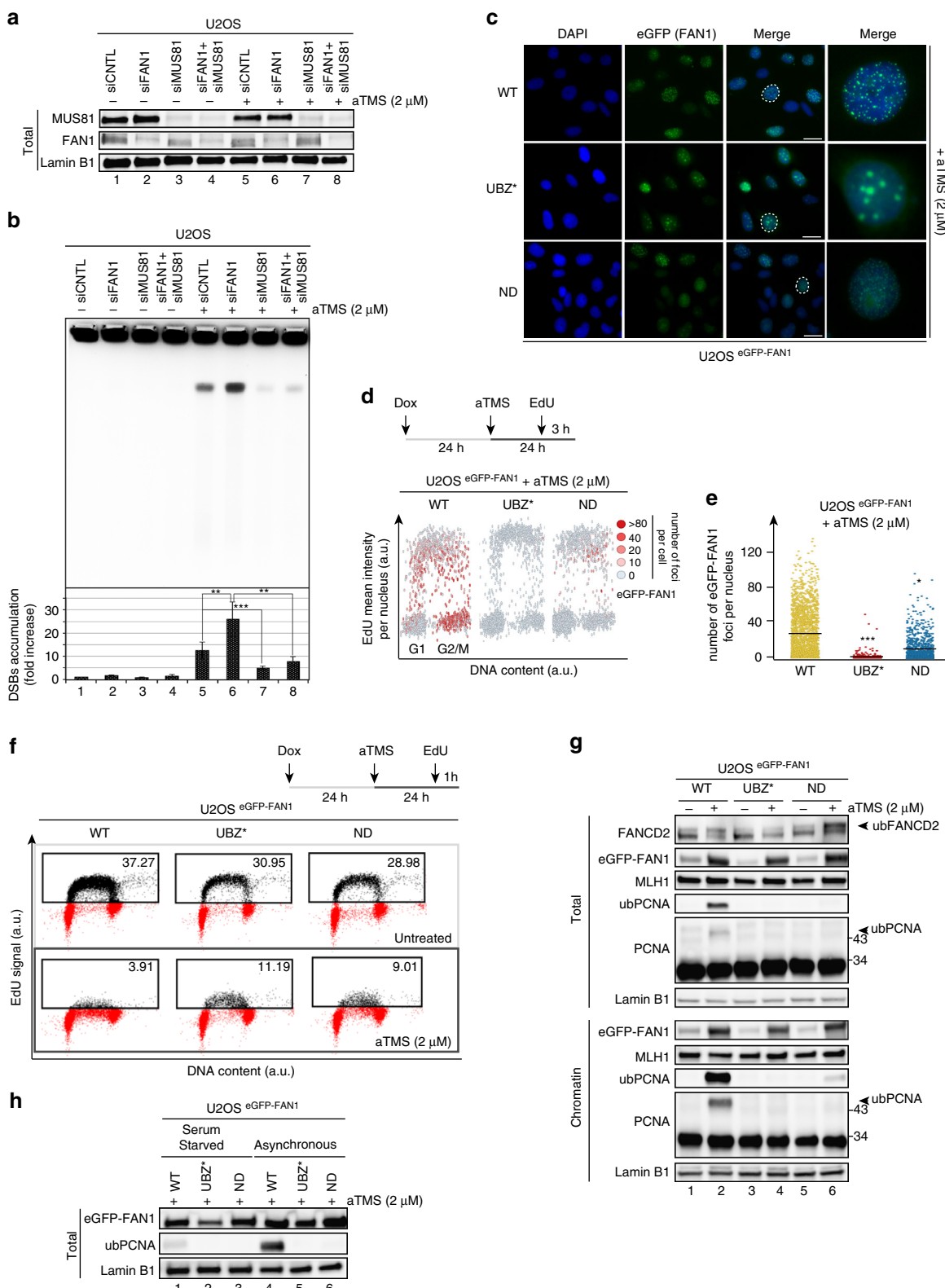

Replication fork arrest is accompanied by monoubiquitylation of FANCD2. As shown in Fig. 1g, total extracts of aTMS-treated eGFP-FAN1 WT and eGFP-FAN1 ND cell lines contained substantial levels of ub-FANCD2, while the protein was largely unmodified in the U2OS eGFP-FAN1 UBZ* cells. Because fork arrest will also cause an exchange of replication factors for repair proteins, we studied the aTMS-induced changes in chromatin composition of our U2OS cell lines. Interestingly, in addition to FAN1 accumulation, we also observed that the chromatin of eGFP-FAN1 WT cells—but not that of the other lines—contained large amounts of monoubiquitylated PCNA (Fig. 1g and Supplementary Fig. 1e). As PCNA ubiquitylation typically occurs when the progression of replicative DNA polymerases is hindered by obstacles in template DNA, our finding indicates that aTMS-stabilised G4s hinder replication fork progression. Taken together with the finding that the aTMS-induced eGFP-FAN1 WT foci formed predominantly in S and G2 phases of the cell cycle (Fig. 1d), and that U2OS cells expressing eGFP-FAN1 WT incorporated only very limited amounts of EdU into their DNA (Fig. 1f), our findings imply that FAN1 is involved in controlling the progression of the replication machinery through these lesions. This notion was supported by the finding that ub-PCNA levels were significantly reduced in aTMS-treated serum-starved cells (Fig. 1h) most of which were in the G0/G1 stages of the cell cycle.

ub-PCNA helps recruit specialized polymerases that catalyse translesion synthesis (TLS) as part of the DNA damage tolerance pathway[19]. In line with this finding, the TLS DNA polymerase polη that acts in concert with ub-PCNA also accumulated in the chromatin of eGFP-FAN1 WT cells upon G4s stabilization. In contrast, another TLS polymerase, REV1, appeared to be excluded from the chromatin of aTMS-treated cells (Supplementary Fig. 1f).

**FAN1 localization to aTMS-induced foci requires ub-PCNA.** Because FAN1 recruitment to MMC-induced foci was shown to be dependent on FANCD2[2–5], we wanted to test whether the same was true also for its recruitment to G4s and whether FANCD2 status affected PCNA ubiquitylation. eGFP-FAN1 co-localized with only a small subset of FANCD2 foci in the aTMS-treated cells (Supplementary Fig. 2a). siRNA-mediated FANCD2 depletion caused only a slight reduction of PCNA ubiquitylation (Fig. 2a) and affected the number (Fig. 2b, c and Supplementary Fig. 2a) and the intensity (Fig. 2b, d and Supplementary Fig. 2a, b) of the eGFP-FAN1 foci to only a limited extent. To confirm that the aTMS-induced FAN1 foci formed at blocked replication forks, we examined their co-localisation with PCNA. In eGFP-FAN1 WT cells, FAN1 foci mostly co-localised with those formed by PCNA (Supplementary Fig. 2c) or ub-PCNA (Fig. 2b) and this

situation was largely unaffected by knock-down of FANCD2 (Fig. 2a, b). In contrast, knock-down of RAD18 (Fig. 2a), the ubiquitin E3 ligase that modifies PCNA[20–23], caused diffusion of PCNA throughout the nucleus (Supplementary Fig. 2c) and abrogated both the number (Fig. 2b, c and Supplementary Fig. 2c) and the intensity (Fig. 2b, d and Supplementary Fig. 2b, c) of the eGFP-FAN1 foci. Taken together, the above evidence implies that FANCD2 does not act upstream of FAN1 in response to aTMS. We therefore argued that the FAN1 UBZ domain might be required for docking with another ubiquitylated polypeptide bound at G4s.

As anticipated, RAD18 knock-down abrogated PCNA ubiquitylation. Unexpectedly, it caused a significant decrease in the number of FAN1 foci in aTMS treated cells. It also affected the number of foci under unperturbed conditions to a greater extent than FANCD2 deficiency (Supplementary Fig. 2d–f). RAD18 has been implicated in FANCD2 ubiquitylation[24, 25], however, the major substrate and distal effector of RAD18-mediated ubiquitylation in DNA damage response is PCNA[26]. This fact and our observations suggest that it is primarily ub-PCNA and only to a lesser extent ub-FANCD2 that recruits FAN1 to G4s. Should this be the case, then FAN1 accumulation at G4 sites should depend on PCNA ubiquitylation. This hypothesis was substantiated in U2OS cells stably expressing a Strep-HA tagged PCNA K164R variant refractory to RAD18-dependent modification. In these cells, both FAN1 and PCNA failed to form subnuclear foci following aTMS treatment (Fig. 2e, f). Moreover, the distinctive aTMS-induced pattern of FAN1- but not FANCD2 foci (Supplementary Fig. 2g–i), was altered in cells treated with T2 amino alcohol (T2AA) (Fig. 2g–j), a small molecule that blocks the association between ub-PCNA and its interaction partners[27].

**FAN1 specifically binds to ub-PCNA via its UBZ domain.** To test whether FAN1 and PCNA interact, we incubated GST-tagged FAN1 with purified recombinant His-tagged ub-PCNA. As shown in Fig. 3a, GST-FAN1 specifically and effectively captured ub-PCNA at concentrations above ~10 nM (Supplementary Fig. 3a), whereas GST-FAN1 UBZ* failed to do so (Fig. 3a). His-ub-PCNA pull-down experiments using lysates of U2OS cells expressing eGFP-FAN1 confirmed the UBZ-dependent interaction between FAN1 and ub-PCNA (Supplementary Fig. 3b). Moreover, eGFP antibody beads retrieved both modified and unmodified forms of PCNA from extracts of aTMS-treated cells expressing eGFP-FAN1 WT, but not eGFP-FAN1 UBZ* (Fig. 3b). In line with this finding, interaction between FAN1 and ub-PCNA was significantly reduced in cells depleted of RAD18 (Fig. 3c). Taken together, these data indicate that FAN1 directly interacts with the ubiquitylated form of PCNA and that this interaction requires its UBZ domain.

**Fig. 1** FAN1 prevents fork collapse and limits DNA synthesis by promoting PCNA ubiquitylation. **a** Immunoblot of extracts of U2OS cells transfected with siRNAs against FAN1 and/or MUS81, and treated or mock-treated with aTMS (2 μM; 48 h). The antibodies used are shown on the left. The figure shows a representative blot of three independent experiments. **b** PFGE of DNA isolated from cells treated as described in **a** was performed to visualize DSB induction. A quantification of three independent experiments is shown. Data are represented as mean ± s.d. ($n = 3$). Statistical analysis was carried out using unpaired, two-tailed $t$-tests. $P$ values expressed as **($P < 0.01$) or ***($P < 0.001$) were considered significant. **c** Representative images of U2OS cells expressing the indicated eGFP-FAN1 variants after treatment with aTMS (2 μM; 24 h). Scale bar: 25 μm. **d** Quantitative image-based cytometry (QIBC) of eGFP-FAN1 foci in cells from **c** pulse-labeled with EdU during the last 3 h of aTMS treatment. The heat map indicates the mean eGFP-FAN1 intensity per nucleus. **e** Quantification of eGFP-FAN1 foci count derived from the QIBC analysis in **d**. Median levels are indicated by black bars. Statistical analysis was carried out using unpaired, two-tailed $t$-tests. $P$ values expressed as ***($P < 0.001$) and *($P < 0.05$) were considered significant, $n = 3$. **f** Cells as in **c** were treated with EdU (1 h) and Click chemistry. EdU incorporation was evaluated by FACS. **g** Total cell extracts and chromatin-enriched fractions of cells as in **c** were analysed by immunoblotting using the antibodies shown on the left. A representative blot of four independent experiments is shown. **h** Immunoblot analysis of total cell extracts of exponentially-growing or serum-starved U2OS cells expressing the indicated eGFP-FAN1 variants following treatment as in **c**. A representative blot of two independent experiments is shown

**FAN1 PIP-box confers specificity to ub-PCNA interaction**. Proteins interacting with PCNA generally contain a conserved sequence motif known as PCNA-interacting peptide or PIP box (Supplementary Fig. 4a), and polypeptides interacting with ub-PCNA contain in addition also a ubiquitin-binding domain (UBD). We therefore searched for the presence of a PIP box in human FAN1. Sequence alignments identified a potential non-canonical PIP box sequence located immediately upstream of the

UBZ domain that was highly conserved in evolution (Fig. 4a). To test whether this motif was functional, we mutated I30 and F34 to alanines (Fig. 4a) and studied the ability of this GST-FAN1 PIP* variant to interact with PCNA and ub-PCNA in GST pull-down experiments. As shown in Fig. 4b, GST-FAN1 WT, GST-FAN1 PIP* or GST-FAN1 UBZ* did not interact with unmodified PCNA at the tested concentrations. In contrast, GST-FAN1 WT bound ub-PCNA with high affinity, while neither GST-FAN1

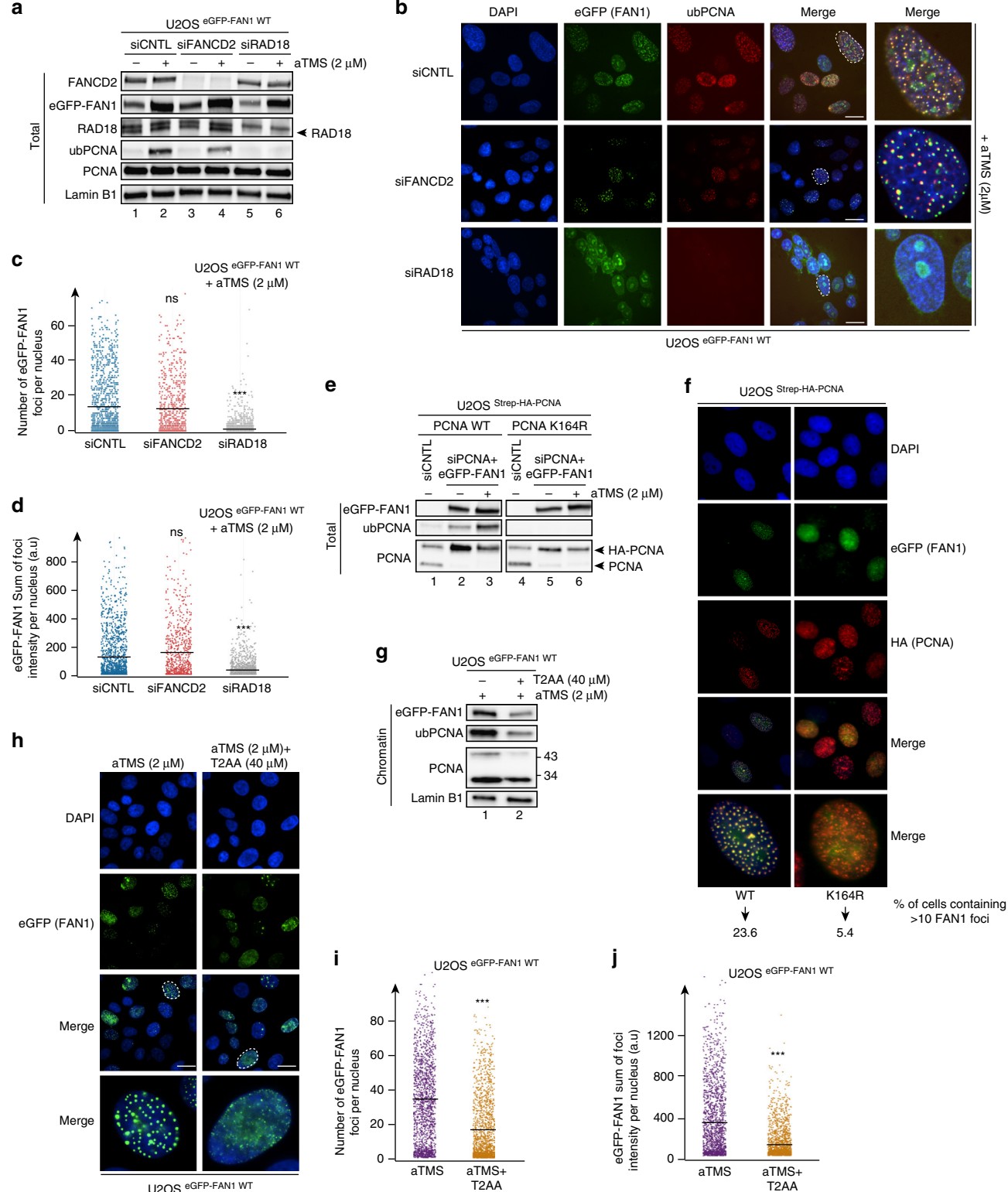

PIP* nor GST-FAN1 UBZ* did so. This indicated that the PIP box and the UBZ domain of FAN1 function in concert to bind ub-PCNA.

In order to substantiate these findings in vivo, we generated a stable U2OS cell line expressing Dox-inducible eGFP-FAN1 PIP* (Supplementary Figs. 1b and 4b). aTMS treatment failed to induce PCNA ubiquitylation in these cells, in contrast to cells expressing eGFP-FAN1 WT (Fig. 4c). As in the case of the eGFP-FAN1 UBZ* variant, mutation of the PIP box abolished the interaction between eGFP-FAN1 and ub-PCNA in vivo (Fig. 4d) and affected eGFP-FAN1 focus formation upon aTMS treatment (Fig. 4e–g) as well as in unperturbed cells (Supplementary Fig. 4c, d). eGFP-FAN1 PIP* also failed to restrain DNA synthesis as measured by EdU incorporation (Fig. 4h), or prevent fork collapse and DSBs accumulation in aTMS-treated cells (Fig. 4i).

**The FAN1/ub-PCNA interaction helps prevent fork collapse.** The crosstalk between FAN1 and ub-PCNA at G4s was further corroborated in U2OS cells expressing Strep-HA-tagged PCNA WT or its non-ubiquitylatable K164R mutant. Concomitant siRNA-mediated knock-down of endogenous PCNA and FAN1 led to a reduction of aTMS-induced PCNA ubiquitylation in cells expressing the Strep-HA-PCNA WT (Fig. 5a, left panel). As anticipated, no PCNA ubiquitylation was observed in cells expressing Strep-HA-PCNA K164R (Fig. 5a, right panel). As in U2OS cells tested above (Fig. 1b), FAN1 knock-down gave rise to

substantial genomic instability in aTMS-treated cells. Importantly, the observed DSB accumulation was substantially greater in cells expressing Strep-HA-PCNA WT than in cells expressing the Strep-HA-PCNA K164R variant (Fig. 5b). Interestingly, siRNA-mediated knock-down of FAN1 and USP1 (Supplementary Fig. 5), the enzyme responsible for deubiquitylating PCNA, caused similar genomic instability in U2OS cells, but the number of DSBs in the double knock-down was increased approximately two-fold (Fig. 5c), thus indicating that an increase in ub-PCNA may worsen the genomic instability in FAN1 depleted cells.

**Binding to ub-PCNA ensures UV-induced FAN1 foci formation.** G4s form on the single-stranded template strand behind the MCM helicase. We therefore asked whether FAN1 was implicated also in the processing of other template strand lesions, such as UV photodimers. To this end, we irradiated the U2OS cell line expressing eGFP-FAN1 WT with UV (30 J/m$^2$). Like aTMS, UV irradiation caused an increase in the number of FAN1 foci, which mostly co-localised with those formed by PCNA (Fig. 6a). This situation was largely unaffected by knock-down of FANCD2 (Fig. 6a–c), whereas knock-down of RAD18 resulted in a decrease of a both the number (Fig. 6a, b) and the intensity (Fig. 6a, c) of the eGFP-FAN1 foci. Moreover, UV-induced formation of FAN1 foci was strongly reduced in cells treated with T2AA (Fig. 6d, e) and in cells expressing the PIP* and the UBZ* variants of the protein (Fig. 6f–h), thus recapitulating what was observed in

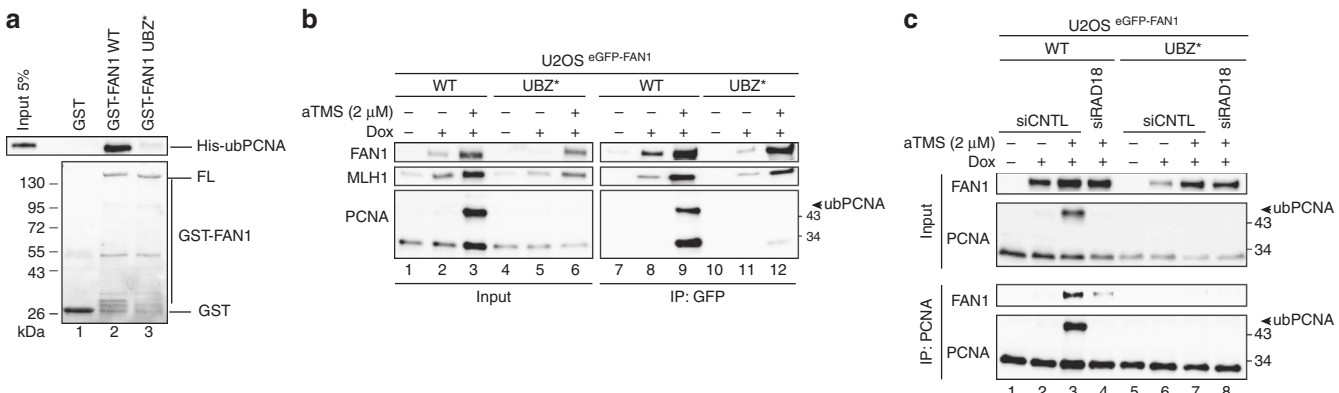

**Fig. 3** FAN1 directly interacts with ub-PCNA through its UBZ domain in vitro and in vivo. **a** GST, full-length GST-FAN1 WT and the UBZ* mutant were incubated with purified ub-PCNA and subsequently retrieved with glutathione beads. ub-PCNA in input and pull-downs was analysed by immunoblotting. **b** Chromatin-enriched fractions derived from U2OS cells inducibly-expressing eGFP-tagged FAN1-WT and the UBZ* mutant, treated or mock-treated with aTMS (2 μM; 24 h), were incubated with anti-eGFP affinity resin. Inputs and immunoprecipitates were analysed with the indicated antibodies. **c** Chromatin-enriched fractions derived from cells as in **b**, treated or mock-treated with aTMS (2 μM; 24 h) and transfected or not with siRNAs against RAD18, were incubated with an anti-PCNA antibody. The immunoprecipitates were analysed with antibodies against PCNA and FAN1

**Fig. 2** FAN1 localization to aTMS-induced foci requires ub-PCNA but not FANCD2. **a** PCNA ubiquitylation estimated by immunoblotting analysis of total extracts derived from U2OS cells expressing eGFP-tagged FAN1 WT and transfected with the indicated siRNAs, treated or mock-treated with aTMS (2 μM; 24 h). A representative blot of four independent experiments is shown. **b** Cells as in **a** were transfected with the indicated siRNAs and treated with aTMS (2 μM; 24 h) before immunostaining with anti-monoubiquityl-PCNA (K164) antibody. Representative images are shown. Scale bar: 25 μm. **c**, **d** Quantification of eGFP-FAN1 foci count (**c**) and the sum of their intensities (**d**) was derived from QIBC analysis of **b**. Median levels are indicated by black bars. Statistical analysis was carried out using unpaired, two-tailed t-tests. P values expressed as ***(P < 0.001) were considered significant, n = 3. **e**, **f** U2OS cells stably-expressing Strep-HA-PCNA WT or the K164R mutant were transiently transfected with eGFP-FAN1 WT plasmid before exposure to aTMS (2 μM; 24 h). Immunoblotting analysis of their extracts (**e**) was carried out with the antibodies shown on the left. A representative blot of two independent experiments is shown. Immunostaining (**f**) was done with an anti-HA antibody. Representative images are shown. Scale bar: 25 μm. **g**, **h** Cells as in **a** were treated with aTMS (2 μM; 24 h) and incubated or not with T2AA (40 μM; 6 h). Immunoblotting analysis (**g**) was carried out with the antibodies shown on the left. **h** Immunostaining of eGFP-FAN1 was performed. **h** shows representative images. **i**, **j** Quantification of eGFP-FAN1 foci count (**i**) and the sum of their intensities (**j**) derived from the QIBC analysis of **h**. Median levels are indicated by black bars. Statistical analysis was carried out using unpaired, two-tailed t-tests. P values expressed as ***(P < 0.001) were considered significant, n = 3

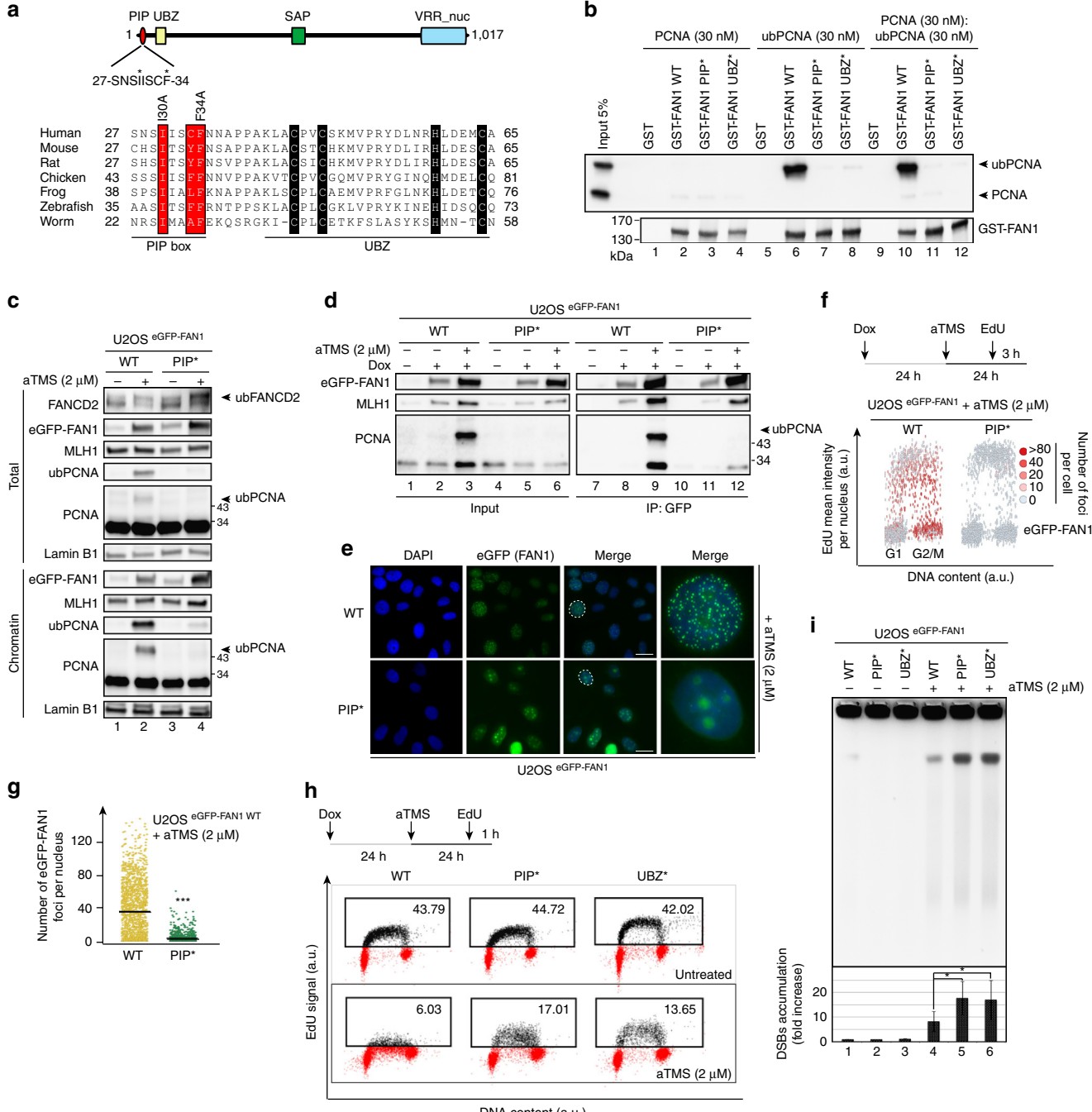

**Fig. 4** A non-canonical PIP-box motif and the UBZ domain of FAN1 are both required for binding to ub-PCNA. **a** Schematic representation of human FAN1 (top). Sequence alignment of the non-canonical PIP-box of FAN1 from different species. Residues important for ub-PCNA binding are highlighted in red. The CCHC residues of the UBZ domain zinc finger are highlighted in black. **b** GST, GST-FAN1 WT, GST-FAN1 PIP* and GST-FAN1 UBZ* variants were incubated with purified His-PCNA or His-ub-PCNA and retrieved with glutathione beads. PCNA and ub-PCNA in input and pull-downs were analysed by immunoblotting. A representative blot of three independent experiments is shown. **c** Total extracts and chromatin-enriched fractions of U2OS cells with the indicated genotypes treated with aTMS (2 μM; 24 h) were analysed by immunoblotting using the indicated antibodies. A representative blot of three independent experiments is shown. **d** Chromatin-enriched fractions derived from cells as in **c**, treated or mock-treated with aTMS (2 μM; 24 h), were incubated with anti-eGFP affinity resin. Inputs and immunoprecipitates were analysed by immunoblotting with the indicated antibodies. A representative blot of four independent experiments is shown. **e** Representative images of eGFP-FAN1 foci in cells as in **c**. Scale bar: 25 μm. **f** QIBC of eGFP-FAN1 foci was performed in cells as in **c** exposed to aTMS (2 μM; 24 h) and pulse-labeled with EdU during the last 3 h of aTMS treatment. The heat map indicates the mean eGFP-FAN1 intensity per nucleus. **g** Quantification of eGFP-FAN1 foci count derived from the QIBC analysis in **f**. Median levels are indicated by black bars. Statistical analysis was carried out using unpaired, two-tailed t-tests. P values expressed as ***(P < 0.001) were considered significant, n = 3. **h** U2OS cells with the indicated genotypes were left untreated or treated with aTMS (2 μM; 24 h), followed by incubation with EdU (1 h) and Click chemistry. EdU incorporation was evaluated by FACS. **i** DSBs formation was evaluated by PFGE in same cells as in **h** treated or mock-treated with aTMS (2 μM; 48 h). Quantification of three independent experiments is shown. Data are represented as mean ± s.d. (n = 3). Statistical analysis was carried out using unpaired, two-tailed t-tests. P values expressed as *(P < 0.05) were considered significant

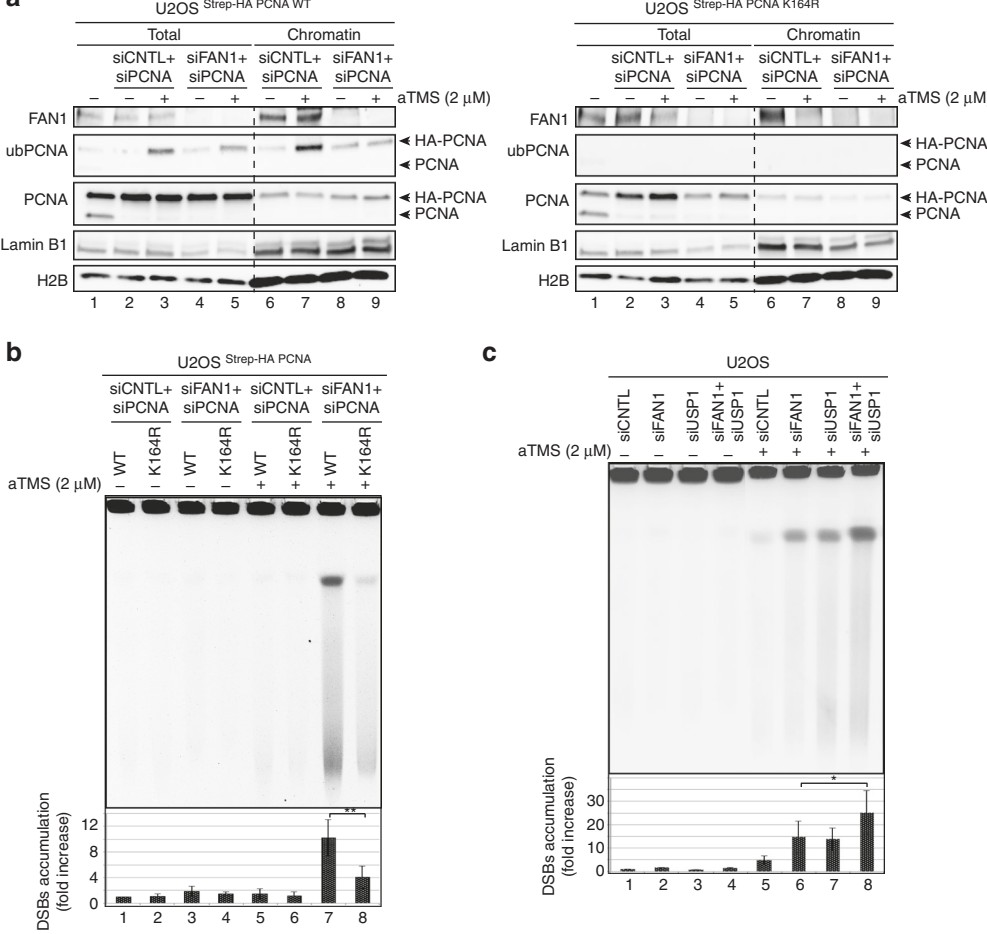

**Fig. 5** The functional interaction between FAN1 and ub-PCNA is required for preventing fork collapse. **a** U2OS cells stably-expressing Strep-HA-PCNA WT (left panel) or the K164R mutant (right panel) were transfected with the indicated siRNAs and treated or mock-treated with aTMS (2 μM; 24 h). Total cell extracts and chromatin-enriched fractions were analysed by immunoblotting with the indicated antibody. A representative blot of two independent experiments is shown. **b** DNA isolated from cells as in **a** was subjected to PFGE to visualize DSB induction upon treatment with aTMS (2 μM; 24 h). Quantification of three independent experiments is shown. Data are represented as mean ± s.d. (n = 3). Statistical analysis was carried out using unpaired, two-tailed *t*-tests. *P* values expressed as **(*P* < 0.01) were considered significant. **c** DNA isolated from U2OS cells transfected with siRNAs against FAN1 and/or USP1, and treated or mock-treated with aTMS (2 μM; 48 h) was subjected to PFGE to visualize DSB induction. A quantification of three independent experiments is shown. Data are represented as mean ± s.d. (n = 3). Statistical analysis was carried out using unpaired, two-tailed *t*-tests. *P* values expressed as *(*P* < 0.05) were considered significant

aTMS-treated cells. In contrast, UV-induced FANCD2 foci were altered only to a limited extent (Supplementary Fig. 6a–c) upon exposure to T2AA, thus reinforcing the notion that FANCD2 does not operate upstream of FAN1 at halted forks. Along the same line, mutation of the PIP or the UBZ domain substantially reduced the extent of PCNA ubiquitylation in the irradiated cells (Fig. 6i).

**MMC-induced FAN1 foci form independently of its PIP-box.** A similar scenario was recapitulated in cells exposed to brief HU treatment, where FAN1 mutations in the PIP or UBZ domain abrogated focus formation (Supplementary Fig. 7a–d). We also explored the role of the FAN1 interaction with ub-PCNA in cellular response to MMC (Fig. 7a), that blocks replication fork progression by covalently linking the Watson and the Crick strands, a mode of action different from aTMS, UV or HU. Surprisingly, the eGFP-FAN1 PIP* variant, but not eGFP-FAN1 UBZ*, retained the ability to form subnuclear foci in treated cells (Fig. 7b–d) and to bind ub-FANCD2 (Fig. 7e). This shows that FAN1 recruitment to MMC-arrested forks does not require PCNA ubiquitylation. Taken together, the above evidence

implicates FAN1 in the processing of arrested replication forks, but clearly demonstrates that its mode of action varies depending on the type of barrier it encounters.

**FAN1/ub-PCNA interaction stabilizes BRCA2-deficient cell DNA.** Given that knock-down of FAN1 gave rise to substantial genomic instability following aTMS treatment, we asked whether this instability arose as a consequence of rescue of collapsed replication forks by BRCA2-assisted homologous recombination (HR)[28]. We therefore knocked-down BRCA2 with siRNA (Fig. 8a) and examined the genetic stability of the aTMS-treated cells by PFGE. As shown in Fig. 8b, BRCA2 knock-down did not significantly increase the number of DSBs when compared to the control treated cells. In contrast, knock-down of both FAN1 and BRCA2 caused a substantial increase in DSB number, which was reflected also in increased cell death (Fig. 8c–e). The finding that FAN1 and BRCA2 are not epistatic in the processing of replication forks halted at G4s implies that the former protein is likely involved in controlling the restart and stability of blocked replication forks rather than in HR. However, BRCA2 deficiency causes replication stress and has been reported to lead to an

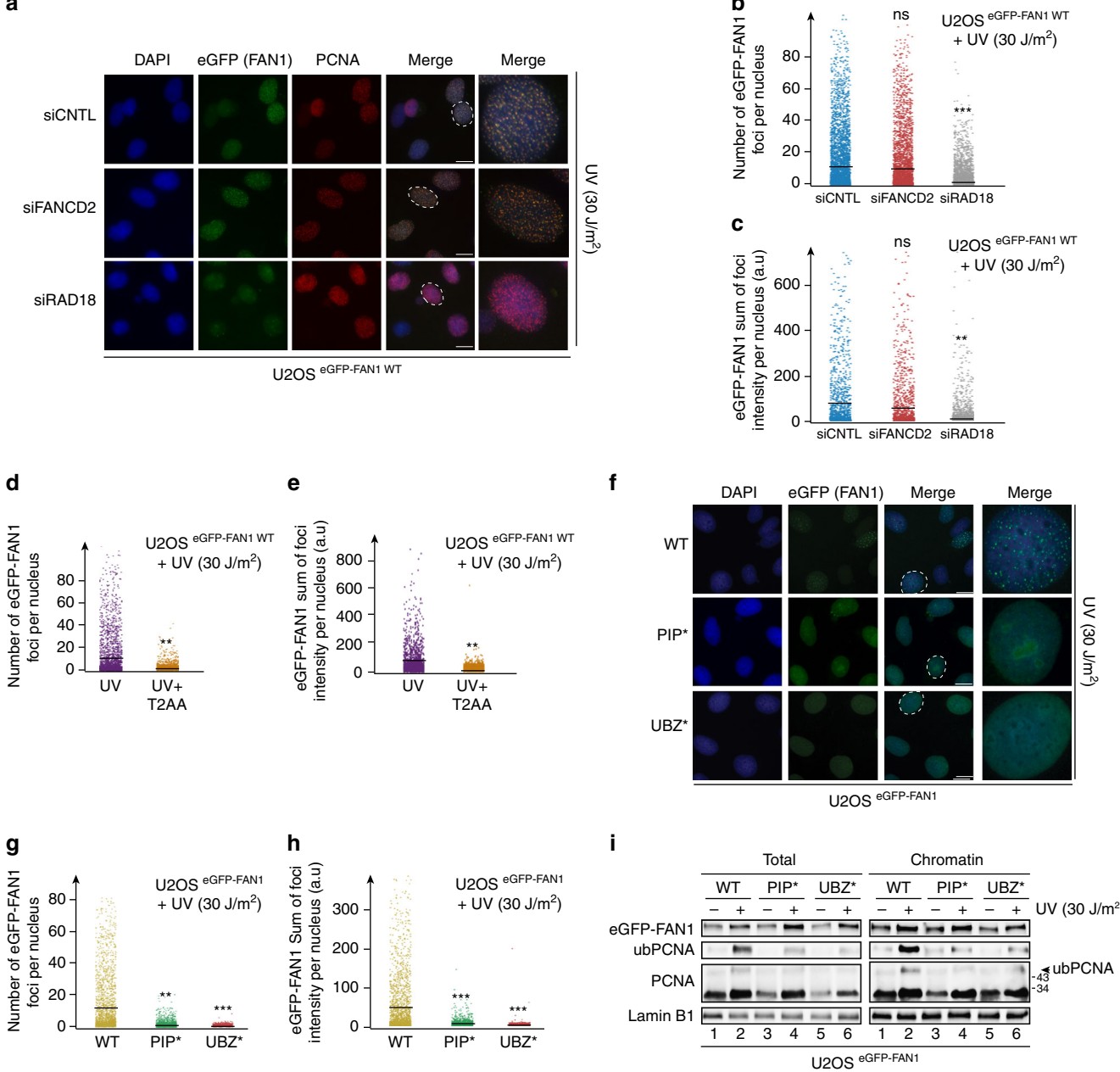

**Fig. 6** The FAN1 PIP-box motif is required for FAN1 foci formation upon UV exposure. **a** U2OS cells expressing eGFP-tagged FAN1 WT were transfected with the indicated siRNAs and treated or mock-treated with UV (30 J/m², 4 h release) before immunostaining with anti-PCNA antibody. Representative images are shown. Scale bar: 25 μm. **b, c** Quantification of eGFP-FAN1 foci count (**b**) and the sum of their intensities (**c**) derived from QIBC analysis of **a**. Median levels are indicated by black bars. Statistical analyses were carried out using unpaired, two-tailed t-tests. P values expressed as ***(P < 0.001) and **(P < 0.01, n = 3) (**d, e**) Cells as in **a** were treated with UV (30 J/m², 4 h release) and incubated or not with T2AA (40 μM; 6 h). Quantification of eGFP-FAN1 foci count (**d**) and the sum of their intensities (**e**) was obtained from QIBC analysis. Median levels are indicated by black bars. Statistical analysis was carried out using unpaired, two-tailed t-tests. P values expressed as **(P < 0.01) were considered significant (n = 3). **f** Representative images of U2OS cells expressing the indicated eGFP-FAN1 variants after treatment with UV (30 J/m², 4 h release). Scale bar: 25 μm. **g, h** Quantification of eGFP-FAN1 foci count (**g**) and the sum of their intensities (**h**) was obtained from the QIBC analysis of **f**. Median levels are indicated by black bars. Statistical analyses were carried out using unpaired, two-tailed t-tests. P values expressed as ***(P < 0.01) or **(P < 0.01) were considered significant, n = 3. **i** Total cell extracts and chromatin-enriched fractions of U2OS cells expressing the indicated eGFP-FAN1 variants and treated or mock-treated with UV (30 J/m², 4 h release) were analysed by immunoblotting using the indicated antibodies. A representative blot of four independent experiments is shown

increased use of the TLS pathway following HU treatment[29]. Based on data presented above, we postulated that the ability of FAN1 to sustain or even augment PCNA ubiquitylation might facilitate TLS and attenuate genomic instability in BRCA2-deficient cells. We therefore examined the effect of siRNA-mediated BRCA2 knock-down in U2OS cells expressing

eGFP-FAN1 WT, eGFP-FAN1 PIP*, eGFP-FAN1 UBZ* and eGFP-FAN1 ND (Fig. 8f) on chromosomal stability. As shown in Fig. 8g, h, BRCA2-depleted cells expressing the FAN1 mutants displayed enhanced levels of chromosomal abnormality when compared to cells expressing eGFP-FAN1 WT. Moreover, FAN1 WT limits unscheduled DNA synthesis in BRCA2-depleted cells,

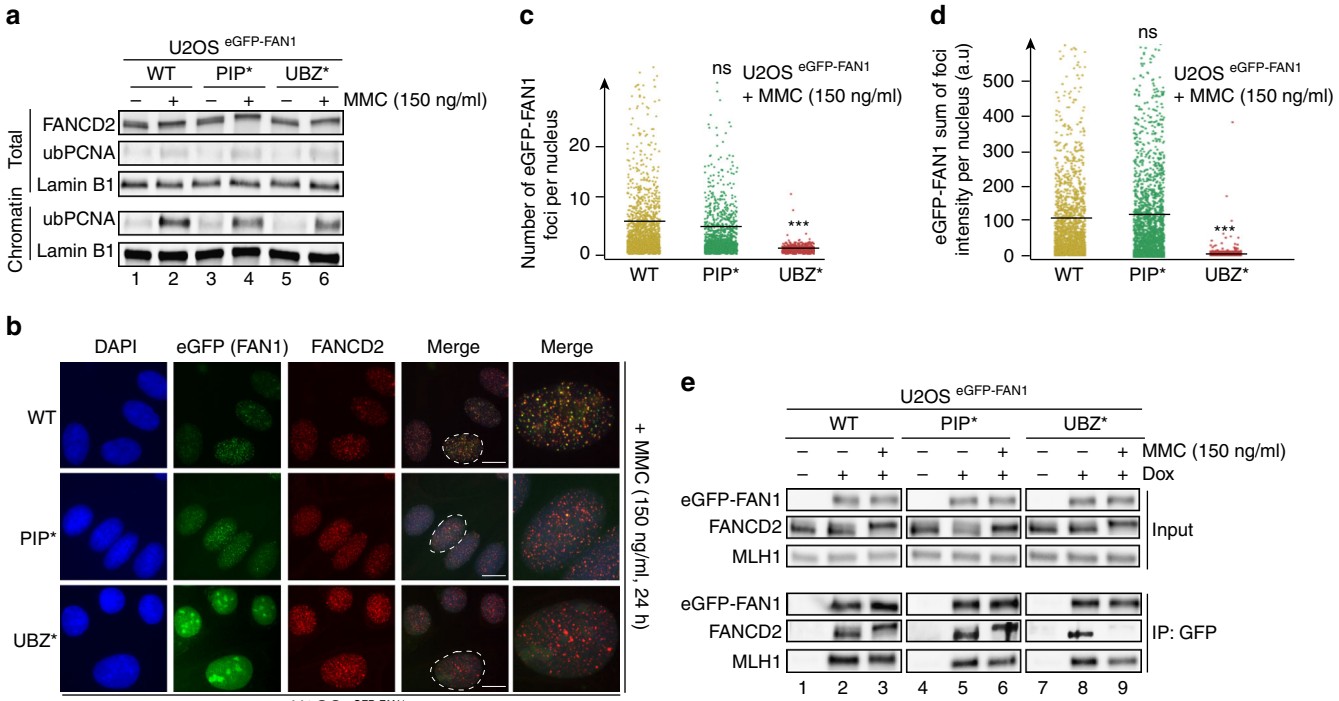

**Fig. 7** The FAN1 PIP-box motif is not required for FAN1 foci formation upon exposure to MMC. **a** Total cell extracts and chromatin-enriched fractions of U2OS cells expressing the indicated eGFP-FAN1 variants, treated or mock-treated with MMC (150 ng/ml, 24 h), were analysed by immunoblotting using the indicated antibodies. A representative blot of three independent experiments is shown. **b** Cells as in **a** were immunostained with anti-FANCD2 antibody. Representative images are shown. Scale bar: 25 μm. **c, d** Quantification of eGFP-FAN1 foci count (**c**) and the sum of their intensities (**d**) was obtained from QIBC analysis of **b**. Median levels are indicated by black bars. Statistical analyses were carried out using unpaired, two-tailed $t$-tests. $P$ values expressed as ***($P < 0.01$) were considered significant, $n = 3$. **e** Total cell extracts derived from cells as in **a**, treated or mock-treated with MMC (150 ng/ml, 24 h), were incubated with anti-eGFP affinity resin. Inputs and immunoprecipitates were analysed by immunoblotting with the indicated antibodies. A representative blot of two independent experiments is shown

whereas the PIP*, UBZ* and ND variants, did so to a much more limited extent (Supplementary Fig. 8). Taken together, these findings suggest that FAN1, and specifically its interaction with ub-PCNA, might be required to counteract elevated replication stress levels occurring in BRCA-mutated tumors, similarly to FANCD2[30, 31].

## Discussion

*FAN1* knock-out DT40 cells[7] and cell lines in which the protein was knocked-down[2–5] were reported to be hypersensitive to cisplatin and MMC. Because recruitment of FAN1 to nuclear foci induced by the latter reagents appeared to be dependent on FANCD2, search for the biological role of FAN1 has focused primarily on the metabolism of extrinsically-induced ICLs and on its anticipated link between FAN1 and FA. Although its involvement in ICL processing is beyond doubt, the absence of *FAN1* mutations in FA and the link to KIN[10–12], suggested that the protein has another function. This notion was further supported by the finding that expression of FAN1 variants either mutated in[13] or lacking[11] the UBZ domain rescued the sensitivity of FAN1-deficient cells to MMC. Recent experimental evidence points to a role of FAN1 in the metabolism of arrested or hindered replication forks, as exemplified by genomic instability arising when DNA synthesis is inhibited by aphidicolin[14] or HU[13]. Aphidicolin and HU block the synthesis of the template strand through interfering with the incorporation of deoxycytidine or through depletion of the nucleotide pool, respectively. We set out to test whether barriers on the template strand are subject to similar criteria. Here, we demonstrate that FAN1

directly interacts with ub-PCNA at replication forks stalled at secondary structures on the template strand and that this interaction requires both the UBZ domain and the newly-identified non-canonical PIP box motif of FAN1 (Fig. 4e). This is reminiscent of recruitment of Y-family TLS polymerases to bulky lesions such as UV photodimers, or of SNM1A to stalled replication forks[32]. Interestingly, the latter nuclease can partially compensate for FAN1 deficiency in ICL repair[11], which suggests that the two proteins have a similar *modus operandi*.

The UBZ domain and the PIP box of FAN1 are evolutionarily conserved from *C. elegans* to man (Fig. 4a), which suggests that they may have co-evolved to cope with the increased genomic complexity of higher organisms. While the PIP motif is not required for FAN1 recruitment to MMC-induced damage (Fig. 7b–d), it is needed to elicit a full function of the protein when challenged by different lesions and structures that hinder replication progression. Likewise, the UBZ domain has hitherto been believed to be required solely for the interaction with ub-FANCD2, but we show that it is also needed for ub-PCNA binding and it is possible that it helps to mediate interactions with other, as yet unidentified, ubiquitylated proteins. We therefore posit that FAN1 is a general factor recruited to a variety of structures that block replication forks. As already discussed, forks can be arrested by ICLs, by the lack of nucleotides or by polymerase inhibitors, but they can also be hindered by other lesions, ranging from abasic sites, modified bases or secondary structures to breaks in the template strand. It is most likely that these lesions will be processed by different sets of factors. When ub-PCNA was discovered to recruit several different TLS polymerases to replication-blocking lesions, it was likened to a tool belt that

carried many tools, because it could not anticipate the kind of lesion that awaited it. FAN1 may have a similar role; its PIP motif and UBZ domain provide it with a flexibility necessary to engage in the processing of different lesions, and extend its range by combining its enzymatic activity with multifunctional recruiting factors such as ub-PCNA and/or ub-FANCD2. But these

interactions could also serve to restrict the access of FAN1 to certain substrates. As a nuclease that can cleave several different DNA structures in vitro[2, 4, 6, 8, 9], its activity in vivo needs to be closely controlled. Thus, ICL-arrested forks must be incised either prior to[33, 34] or after TLS[35], and processing of replication forks arrested through nucleotide pool depletion or polymerase

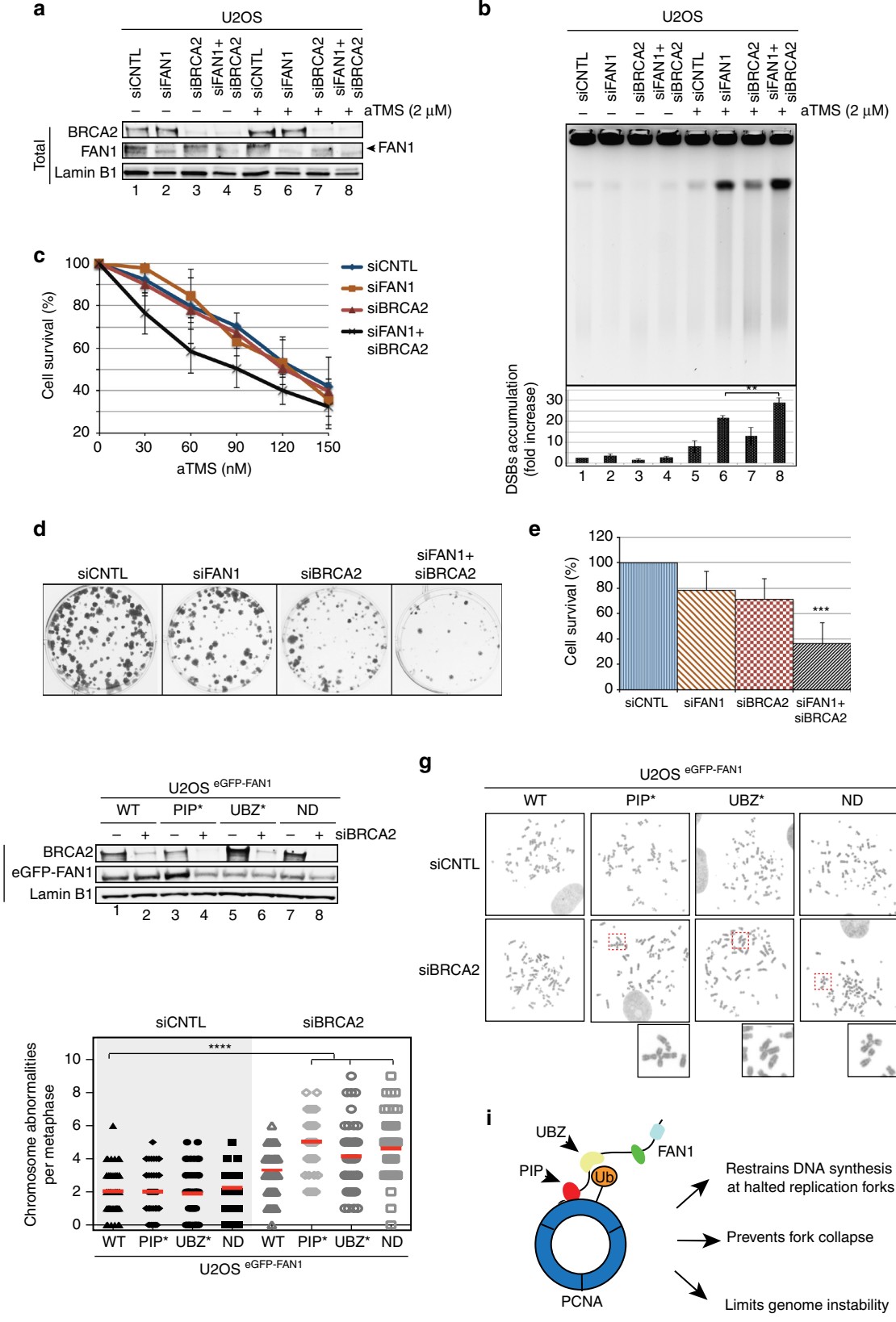

inhibition on the nascent strand may require different cofactors than the release of forks blocked by e.g. G4s on the template strand. The control of the nucleolytic activity is essential, as inappropriate incision of some lesions could cause genomic instability rather than protect against it.

FAN1 is a reader of ub-PCNA but it also enhances PCNA ubiquitylation. It remains to be established whether it does so by stimulating the ubiquitylation machinery, or through inhibiting the rate of PCNA deubiquitylation. It also remains to be established whether the extensive ubiquitylation of PCNA induced by aTMS or UV requires prior DNA cleavage. How FAN1 limits the rate of DNA synthesis when fork progression is hindered [ref. 13 and this work] also requires elucidation. One possibility is that FAN1, together with ub-PCNA, helps channel a subset of stalled forks into the TLS pathway, which utilise polymerases with limited processivity. Support for this prediction comes from biochemical studies showing that FAN1 can traverse an ICL and thus generate a substrate for TLS[8]. However, other pathways, such as template-switching[26, 36–39] or alternative end-joining (AJ)[40–42] might also rely on the activation of the FAN1:ub-PCNA axis.

In our first study of FAN1[2] we postulated that the protein function either in the initial incision step of ICL processing, or in the final, HR-mediated rescue of the collapsed replication fork. Our biochemical evidence showing that FAN1 can traverse ICLs[8] suggests that the primary function of FAN1 lies in the incision of blocked forks. The notion is further supported by the current finding that FAN1 and BRCA2 are not epistatic, which implies that the intermediates generated by FAN1 do not require BRCA2 for resolution. This may be linked to the fact that FAN1 is endowed not only with a 5′-flap endonuclease function, but also with a 5′ to 3′ exonuclease that could generate stretches of ssDNA that are channelled into a pathway distinct from BRCA2-dependent HR. FAN1-mediated processing of blocked replication forks thus appears to be mechanistically distinct from the FA pathway, in which FANCD2 and BRCA2 (FANCD1) cooperate[43]. How these two pathways differ on the molecular level remains to be elucidated. But if intermediates incised by FAN1 are first channelled into TLS rather than HR, this would activate RAD18-mediated PCNA ubiquitylation. Ub-PCNA would then serve as a docking site for the recruitment of factors such as ZRANB3[44], which are indispensable for the resumption of DNA synthesis.

In conclusion, our data demonstrate that the interaction with ub-PCNA is a key facet that regulates FAN1 function and confirm that its role outside of the FA pathway is important for genome maintenance (Fig. 8i). The synthetic toxicity observed in the FAN1/BRCA2 double knock-down cells and the aggravated genomic instability of BRCA2-depleted cells expressing the FAN1 PIP*, UBZ* or ND mutants suggests that inhibition of FAN1 function in BRCA-deficient cancers may cause cell lethality. This may represent a novel cancer therapeutic opportunity that requires further study.

## Methods

**Cell culture**. Human osteosarcoma U2OS cells were grown in Dulbecco's modified Eagle's medium (DMEM) supplemented with 10% foetal calf serum (FCS), 100 U/ml penicillin, and 100 μg ml streptomycin. Using Fugene (Roche) transfection according to the manufacturer's protocol, and hygromycin B (250 μg ml$^{-1}$) and puromycin (1 μg ml$^{-1}$) for selection, we established U2OS T-REx™ cell lines inducibly expressing shRNA-resistant forms of eGFP-FAN1: FAN1-WT (full-length wild-type protein), FAN1-PIP* (I30A/F34A) (full length protein mutated in the non-canonical PIP motif), FAN1-UBZ* (C44A/C47A) (full length protein mutated in the UBZ domain), FAN1-ND (D960A/K977A) (full-length FAN1 mutated in the catalytic endonuclease motif) cloned into the pAIO vector[18] carrying an N-terminal eGFP tag. The sequence of FAN1 was altered by site-directed mutagenesis to become resistant to si/shFAN1 (5′-GUA AGG CUC UUU CAA CGU A-3′). To induce shRNA expression, cells were treated with 0.1 μg ml$^{-1}$ doxycycline (Dox) as indicated. U2OS T-Rex cell lines were grown in DMEM supplemented with 10% Tet system-approved FCS, 100 U/ml penicillin, 100 μg ml$^{-1}$ streptomycin, hygromycin B (250 μg ml$^{-1}$) and puromycin (1 μg ml$^{-1}$). Ultimately, the cDNA integrated in the genome of U2OS T-REx cells was amplified by PCR using the primers eGFP forward (5′ATGGTGAGCAAGGGCGAGGAG3′) and FAN1 reverse (5′TTAGCTAAGGCTTTGGCTCTTAGCTCC3′) and analysed for the presence of mutations by deep sequencing. U2OS cell lines stably expressing Strep-HA-PCNA wild-type or K164R mutant were a gift from Niels Mailand (University of Copenhagen) and obtained by selecting U2OS cells transfected with pcDNA4/TO-Strep-HA-PCNA plasmids in a medium containing zeocin (Invitrogen) as described[45]. All cell lines were tested for mycoplasma and were confirmed free of contamination. The cell lines used in this study were not authenticated and are not found in the database of commonly misidentified cell lines that is maintained by ICLAC and NCBI BioSample.

**siRNA transfections**. siRNA oligos were transfected at a final concentration of 20 nM using Lipofectamine RNAiMAX (Invitrogen). For co-depletion experiments, each siRNAs was 10 nM. In the single depletions, the siRNA (10 nM) was co-transcefected with an equal concentration of a non-targeting, luciferase siRNA (CTNL). CNTL, FAN1, BRCA2, FANCD2, RAD18, MUS81 and USP1 siRNAs were purchased from Microsynth and the sequences (5′ to 3′) were as follows: CNTL (CGU ACG CGG AAU ACU UCG A), FAN1 (GUA AGG CUC UUU CAA CGU A), BRCA2 (UUG ACU GAG GCU UGC UCA GUU), FANCD2 (CAG AGU UUG CUU CAC UCU CUA), RAD18 (AUG GUU GUU GCC CGA GGU UAA), PCNA 3′-UTR (AGA AUA AAG UCC AAA GUC A), MUS81 (CAGCCCUGGUGGAUCGAUA), USP1 (TCGGCAATACTTGCTATCTTATT).

**Pulsed field gel electrophoresis (PFGE)**. PFGE was carried out essentially as described previously[46]. In brief, cells were collected by trypsinization, and agarose plugs containing $7.5 \times 10^5$ ($1.5 \times 10^6$ for 2 plugs) cells were prepared. The plugs were then incubated in lysis buffer [100 mM EDTA, 1% (w/v) sodium lauryl sarcosine, 0.2% (w/v) sodium deoxycholate, 1 mg/ml proteinase K] at 37 °C for 72 h and then washed four times in washing buffer (20 mM Tris-HCl (pH 8.0), 50 mM EDTA) for 30 min before loading onto an agarose gel. Electrophoresis was performed for 21 h at 14 °C in 0.9% (w/v) Pulsed Field-Certified Agarose (Biorad) containing TBE-Buffer, using a Biometra Rotaphor apparatus with the following parameters: voltage, 4 V/cm; switch time, 60–240 s; angle, 120°. The gels were stained with ethidium bromide and analysed using a CCD camera (BioRad).

**Immunofluorescence microscopy**. The U2OS cells were cultured on coverslips. Nuclei were pre-extracted by incubating the coverslips with pre-extraction solution (25 mM Hepes pH 7.4, 50 mM NaCl, 1 mM EDTA, 3 mM MgCl$_2$, 300 mM sucrose, 0.5% Triton-X-100) for 5–10 min at 4 °C. Nuclei were then fixed with 4% formaldehyde/PBS for 10 min at room temperature (RT). Membranes were permeabilized with 0.2% Triton X-100 in PBS for 5 min at 4 °C. Samples were blocked with 3% BSA/PBS and incubated with primary antibodies at RT for 2 h and with the appropriate AlexaFluor-488 (anti-mouse A-11001 or anti-rabbit A-21206), −594 (anti-mouse A-21044 or anti-rabbit A-11012) and −647 (anti-mouse A-21239 or

**Fig. 8** The FAN1/ub-PCNA interaction preserves genome integrity in BRCA2-deficient cells. **a** Immunoblot of extracts of U2OS cells transfected with siRNAs against FAN1 and/or BRCA2, and treated or mock-treated with aTMS (2 μM; 48 h). The antibodies used are shown on the left. A representative blot of three independent experiments is shown. **b** PFGE of DNA isolated from cells treated as described in **a** was performed to visualize DSB induction. Quantification of three independent experiments is shown. Data are represented as mean ± s.d. ($n = 3$). Statistical analysis was carried out using unpaired, two-tailed t-tests. P values expressed as ** ($P < 0.01$) were considered significant. **c** Cell survival of cells transfected as in **a** was evaluated by colony formation assays upon exposure to aTMS ($n = 3$). **d** Colony formation assay of untreated U2OS cells transfected as in **a**. **e** Quantification of colony formation observed in **d** ($n = 3$). **f** U2OS cells stably-expressing the indicated FAN1 variant were depleted or not of BRCA2 and their total cell extracts were analysed by immunoblotting with the indicated antibodies. A representative blot of two independent experiments is shown. **g** Metaphase spreads of cells as in **f** were analysed for chromatid breaks and radial structures. Representative images of metaphase spreads are shown. **h** Quantification of chromosomal abnormalities observed in **f**. Fifty spreads were analysed per single experiment per sample. Statistical significance (P value) was calculated using one-way ANOVA (**** $P < 0.0001$). **i** Scheme of the FAN1:ub-PCNA interaction and its biological implications

anti-rabbit A-21245)-conjugated secondary antibodies (1:1000) (Life Technologies). Coverslips were mounted with Vectrashield® (Vector Laboratories) containing DAPI and sealed. Images were acquired on Leica Fluorescence Microscope at 63x magnification. When Click-iT EdU staining was performed, cells were incubated with 10 µM EdU before fixation or pre-extraction, and EdU detection was performed according to the manufacturer's recommendations (C10340, Thermo Fisher Scientific). Some of the immunofluorescence experiments were analysed by Quantitative Image-Based Microscopy (QIBC) as previously described[47] and within one experiment, similar cell numbers were compared for the different conditions (at least 1000 cells).

**Immunoprecipitations and western blot analyses.** Total cell lysates were prepared from exponentially-growing cells harvested by trypsinization. Cell pellets were washed with cold PBS and lysed in SDS lysis buffer (10 mM TrisHCl pH 8, 1% SDS) supplemented with protease and phosphatase inhibitory mixture EDTA free (Roche), 1 mM sodium orthovanadate, 1 mM sodium fluoride] and N-ethylmaleimide (NEM, 20 mM), an irreversible inhibitor of all cysteine peptidases. The lysates were sonicated and cleared by centrifugation at 12,000 x g for 10 min at 4 °C. The soluble material was collected and protein concentrations were determined using the Lowry assay. Chromatin-derived fractions were obtained with the same procedure except that nuclei were pre-extracted by incubating cell pellets with pre-extraction buffer (25 mM Hepes pH 7.4, 50 mM NaCl, 1 mM EDTA, 3 mM MgCl$_2$, 300 mM sucrose, 0.5% Triton-X-100) supplemented with protease and phosphatase inhibitors [1x complete inhibitory mixture EDTA free (Roche), 1 mM sodium orthovanadate, 1 mM sodium fluoride] and N-ethylmaleimide (NEM, 20 mM) for 5–10 min at 4 °C before incubation with SDS lysis buffer. For direct loading on SDS-PAGE, lysates were heated for 5 min at 95 °C. For eGFP co-IP assays, cells were lysed in eGFP-IP buffer (100 mM NaCl, 0.2% NP-40, 1 mM MgCl$_2$, 10% glycerol, 50 mM Tris-HCl, pH 7.5) supplemented with phosphatase inhibitors (1 mM sodium orthovanadate, 1 mM sodium fluoride), protease inhibitors [1x complete inhibitory mixture EDTA free (Roche) and 0.1 mM PMSF], N-ethylmaleimide (20 mM) and incubated with Benzonase (Novagen) for at least 30 min at 4 °C. After Benzonase digestion, the NaCl and EDTA concentrations were adjusted to 200 and 2 mM, respectively, and lysates were cleared by centrifugation. 4 mg of lysates were incubated with eGFP-Trap agarose beads (ChromoTek) in eGFP-IP buffer (with 200 mM NaCl and 2 mM EDTA final concentrations) for 1 h at 4 °C. Beads were then washed five times with eGFP-IP buffer (with 200 mM NaCl and 2 mM EDTA final concentrations), complexes were boiled in SDS sample buffer and analysed by SDS–PAGE followed by immunoblotting. The blots were blocked with 5% non-fat powdered milk in 0.1% TBS-T (0.1% Tween-20 in TBS) for 30 min and incubated overnight at 4 °C with primary antibodies appropriately diluted in 0.1% TBS-T. Membranes were incubated with the appropriate secondary antibodies conjugated with horseradish peroxidase for 1 h at room temperature. Proteins were visualized using the ECL detection system (Western Bright$^{TM}$, Advansta) imaging on a FusionSolo (Witec AG). Original uncropped images of Western blots used in this study can be found in Supplementary Fig. 9.

**GST-Enrichment of FAN1.** Full-length wild-type or mutated variants of human FAN1 were cloned into a pGEX-2TK and expressed in the E. coli strain BL21. Cultures were grown at 37 °C with shaking at 250 rpm. Protein expression was induced at OD$_{600}$ 0.6 by the addition of 1 mM IPTG at 18 °C for 16–18 h using heat-shock (30 s at 42 °C before addition of IPTG). The cells were lysed by sonication in the presence of protease inhibitors and lysozyme. Bacterial pellets were lysed in 20 mM Tri-HCl pH 8, 150 mM NaCl, 0.1% NP-40, 1 mM DTT, 2 mM ZnCl$_2$, 0.1 mM PMSF followed by sonication for 5 min at 72% (50% cycle) and centrifugation for 5 min at 10,000 x g. Supernatants were incubated with Glutathione Sepharose 4 Fast Flow (GE Healthcare). The bound GST-fragments were washed with 10 bead volumes of 20 mM Tris-HCl pH 8, 150 mM NaCl, 10% glycerol, 1 mM DTT, 2 mM ZnCl$_2$ and 0.1 mM PMSF. GST-FAN1 proteins immobilised on Glutathione Sepharose beads were incubated with purified ubiquitylated PCNA (a generous gift from Titia Sixma, Netherlands Cancer Institute) or unmodified PCNA (ThermoFisher).

**Antibodies.** Antibodies to the following proteins were used in this study: sheep α-FAN1 (1:500, a kind gift from John Rouse, University of Dundee), rabbit α-FANCD2 (1:5000, NB100-182, Novus Biologicals), mouse α-PCNA (1:1000, PC10, SantaCruz), rabbit α-ubiquitylated-PCNA (1:1000, D5C7P, Cell Signaling), mouse α-RAD18 (1:500, ab57447, Abcam), rabbit α-MLH1 (1:5000, ab92312, Abcam), rabbit α-Polη (1:1000, ab17725, Abcam), rabbit α-REV1 (1:1000, H-300, SantaCruz), rabbit α-GFP (1:1000, ab290, Abcam), rabbit α-LaminB1 (1:1000, ab16048, Abcam), rabbit α-histone H2B (1:1000, ab1790, Abcam), mouse α−MUS81 (1:1000, M1445, Sigma Aldrich), rabbit α−USP1 (1:1000, Bethyl labs. A301-700A).

**Colony-formation assay.** U2OS cells were depleted of FAN1 and/or BRCA2. 36 hr after transfection with the indicated siRNAs, the cells were seeded and 24 h later treated or not with increasing concentrations of aTMS. Cells were cultured for

10 days at 37 °C. Colonies were stained with a crystal violet/ethanol (0.5/20%) solution and counted.

**Chromosomal instability.** Metaphase spreads were prepared from U2OS cells with the indicated genotypes. In brief, cells were treated with colcemid (Gibco) for 2 h prior to harvesting by trypsinization. The cell pellet was resuspended in 5 ml of a 0.075 M hypotonic potassium chloride solution and incubated for 15 min at 37 °C. For fixation, cells were fixed in methanol/acetic acid (3:1) solution. Fixed cells were resuspended in fixative to the appropriate cell density, spread on glass slides (Menzel-Gläser, ThermoFisher) and stained with DAPI. Fluorescent (DAPI) images were acquired using Leica Fluorescence Microscope at 63x magnification.

**Flow cytometry analysis.** EdU incorporation was analysed using the Click-iT EdU technology (C10424, ThermoFisher) according to manufacturer's instructions. DNA was stained with DAPI (DNA content) and fluorescece was measured on an Attune NxT Flow Cytometer (ThermoFisher Scientific).

**Data availability.** The data that support the findings of this study are available from the corresponding authors upon request.

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

## Acknowledgements

We thank Titia Sixma for the purified recombinant His-ub-PCNA, Neils Mailand for the U2OS cells expressing Strep-HA-PCNA WT and K164R, John Rouse for the human FAN1 antibody, Michelle Schmid and Sara Przetocka for technical assistance, and Marco Gatti and Lorenza Ferretti for helpful discussions. This work was funded by the Swiss National Science Foundation grants no. 310030B-133123 and 31003A- 149989 to J.J., no. 31003A-156023 to A.A.S. and the European Research Council grant "Myriam" (294537) to J.J.

## Author contributions

A.P. and J.J. conceived the project, designed the experiments, and wrote the manuscript. A.P. conducted most of the experiments. S.K. generated the U2OS cell lines expressing eGFP-FAN1 WT and UBZ*, S.B. carried out a subset of the western blotting experiments, M.B. carried out high-content microscopy and quantitative image analysis and J.P. performed a subset of the PFGE experiments. A.S. generated some of the cell lines, made some of the plasmids used in this study and purified the GST-tagged proteins. A.A.S. generated the sequence alignments and contributed to the conceptual development of the project.

## Additional information

**Competing interests:** The authors declare no competing financial interests.

**Change history:** A correction to this article has been published and is linked from the HTML version of this paper.

