## [Peer Review File · Nature Communications]

PEER REVIEW FILE

Reviewers' Comments:

Reviewer #1 (Remarks to the Author):

FAN1 has been studied in the context of inter-strand crosslink repair as the cells are sensitive to this agent and because FAN1 is recruited to inter-strand crosslink (ICL) sites. This recruitment is mediated by the UBZ domain of FAN1 and an interaction with Ub-FANCD2. Surprisingly, however, the recruitment of FAN1 to ICL sites is not needed to suppress ICL sensitivity, yet it is necessary to suppress chromosomal instability following MMC or HU treatment. These findings suggest that FAN1 has multiple functions mediated by distinct domains and that it has other functions in the context of DNA replication.

Here, Jiricny and colleagues demonstrate that FAN1 also plays an important role at forks stalled by secondary DNA structures stabilized by an analog of telomestatin (aTMS) which binds G4 DNAs. aTMS induces PCNA-UB and recruits FAN1 to these sites. This recruitment is mediated by an interaction between Ub-PCNA and the UBZ domain of FAN1 and is needed for the induction of PCNA-UB. This interaction is also needed for FAN1 to restrain replication and prevent DNA break formation in the presence of aTMS. Thus, in response to aTMS, FAN1 reads UB-PCNA instead of Ub-FANCD2. This is an interesting story that provides important molecular insight into the functions of FAN1. It also suggests that FAN1 may have the ability to differentiate between different DNA structures and mount distinct responses. Finally, the study shows that FAN1 acts in cells lacking BRCA2 and is synthetic lethal with BRCA2, suggesting it may be a useful target in these cancers.

This is a revised manuscript addressing the function of poorly understood protein. The experiments are all very well done and the conclusions are supported by the data shown. Previous technical concerns and minor omissions have been corrected in this revision. Moreover, the authors have done a number of additional experiments to extend the the initial studies, which which were somewhat limited in scope.

Reviewer #3 (Remarks to the Author):

This is an excellent paper and now it is highly suitable for publication in Nature Communications, given the thorough reviews.